# SARS-CoV-2 D614G spike mutation increases entry efficiency with enhanced ACE2-binding affinity

Seiya Ozono [1,2,6], Yanzhao Zhang[1,6], Hirotaka Ode [3,6], Kaori Sano [1], Toong Seng Tan [4], Kazuo Imai [5], Kazuyasu Miyoshi [5], Satoshi Kishigami [2], Takamasa Ueno [4], Yasumasa Iwatani [3], Tadaki Suzuki [1] & Kenzo Tokunaga [1✉]

The causative agent of the COVID-19 pandemic, SARS-CoV-2, is steadily mutating during continuous transmission among humans. Such mutations can occur in the spike (S) protein that binds to the ACE2 receptor and is cleaved by TMPRSS2. However, whether S mutations affect SARS-CoV-2 cell entry remains unknown. Here, we show that naturally occurring S mutations can reduce or enhance cell entry via ACE2 and TMPRSS2. A SARS-CoV-2 S-pseudotyped lentivirus exhibits substantially lower entry than that of SARS-CoV S. Among S variants, the D614G mutant shows the highest cell entry, as supported by structural and binding analyses. Nevertheless, the D614G mutation does not affect neutralization by anti-sera against prototypic viruses. Taken together, we conclude that the D614G mutation increases cell entry by acquiring higher affinity to ACE2 while maintaining neutralization susceptibility. Based on these findings, further worldwide surveillance is required to understand SARS-CoV-2 transmissibility among humans.

[1] Department of Pathology, National Institute of Infectious Diseases, Tokyo, Japan. [2] Faculty of Life and Environmental Sciences, University of Yamanashi, Yamanashi, Japan. [3] Clinical Research Center, National Hospital Organization Nagoya Medical Center, Nagoya, Aichi, Japan. [4] Division of Infection and Immunity, Joint Research Center for Human Retrovirus Infection, Kumamoto University Kumamoto, Japan. [5] Self-Defense Forces Central Hospital, Tokyo, Japan. [6] These authors contributed equally: Seiya Ozono, Yanzhao Zhang, Hirotaka Ode. ✉email: tokunaga@nih.go.jp

The outbreak of the coronavirus disease 2019 (COVID-19) caused by severe acute respiratory syndrome coronavirus 2 (SARS-CoV-2)[1–4] has rapidly spread around the globe, affecting more than 180 countries. During the COVID-19 pandemic, SARS-CoV-2 has accumulated mutations throughout viral genes encoding the ORF1a, ORF1b, ORF3, ORF8, nucleocapsid (N), S proteins, etc., some of which are clade-defining (Nextstrain, http://nextstrain.org/ncov, based on the GISAID data, www.gisaid.org). Mutations in the S protein are especially crucial because the S protein is key for the first step of viral transmission, i.e., entry into the cell by binding to the angiotensin-converting enzyme-2 (ACE2) receptor[1], followed by cleavage with transmembrane protease serine 2 (TMPRSS2)[5,6], both of which are abundantly expressed in not only the airways, lungs, and nasal/oral mucosa[7,8] but also the intestines[9]. However, it remains unclear whether S mutations affect SARS-CoV-2 cell entry. In this study, we analyze ACE2/TMPRSS2 usage of SARS-CoV S and SARS-CoV-2 S (referred to hereafter as SARS-S and SARS2-S, respectively), compare cell entry of natural SARS2-S variants, and focus on the D614G variant by investigating its structure, ACE2-binding affinity, and neutralization susceptibility. Our results indicate that the D614G mutation confers increased entry efficiency resulting from enhanced binding affinity for ACE2 with no influence on the antigenicity of the S protein.

## Results

### SARS2-S-mediated cell entry shows a strong dependence on TMPRSS2 expression.

To quantify S-mediated cell entry, we employed a novel normalization system for cell entry, which we have recently established by generating a lentiviral vector harboring a small luminescent peptide tag, HiBiT, allowing us to precisely normalize input virus doses. This system results in enhanced experimental accuracy in a single-round replication assay[10] in which a small but detectable difference in cell entry on a linear scale is critical because such a difference can be amplified in multiple virus replication cycles. By using this system, we first examined the cell entry activity of lentiviruses pseudotyped with SARS-CoV (Urbani strain) S or SARS-CoV-2 (Wuhan-Hu-1 strain) S in cells expressing ACE2, a major receptor for SARS-S[11]. The lentivirus pseudotyped with SARS-S was able to efficiently enter 293T cells expressing ACE2, whereas the SARS2-S-pseudotyped virus entered the cells much less efficiently (Supplementary Fig. 1). This result suggests that the expression of ACE2 alone is insufficient to support cell entry of SARS-CoV-2.

Because accumulated evidence has shown that the expression of TMPRSS2 enhances both SARS-CoV and SARS-CoV-2 infection[5,6], we next tested whether the coexpression of ACE2 and TMPRSS2 could mediate efficient infection of 293T cells with SARS2-S-pseudotyped lentiviruses. Notably, the dual expression of ACE2 and TMPRSS2 markedly facilitated SARS2-S-mediated cell entry while moderately enhancing entry by the SARS-S pseudovirus (Fig. 1a). These results indicate that SARS2-S-mediated entry into cells is more highly dependent on TMPRSS2 coexpression than that of SARS-CoV, suggesting that they may have different cellular tropisms to some extent.

Nevertheless, ~25-fold differences in cell entry between SARS-S and SARS2-S pseudoviruses were observed (Fig. 1a), leading to the hypothesis that SARS2-S might be less compatible with lentiviral particles. Because a large body of literature has shown that lentiviral compatibility of a variety of viral envelopes is determined by their cytoplasmic tail (CT) domains[12–14], we compared the sequences of this domain (as well as the transmembrane (TM) domain) in S proteins and found that SARS2-S has one (alanine-to-cysteine at position 1247) and two

(valine-to-isoleucine at position 1216 and leucine-to-methionine at position 1233) amino acid differences in the CT and TM domains, respectively (Supplementary Fig. 2a). Thus, we created a SARS2-S C1247A mutant and a chimeric SARS2-S harboring the TM/CT domains of SARS-S. All SARS2-S proteins showed comparable levels of cell entry (Supplementary Fig. 2b) and actual virion incorporation (Supplementary Fig. 2c). These results suggest that the significantly lower rate of cell entry of the SARS2-S pseudovirus was not due to the incompatibility between SARS2-S and a lentiviral vector but rather to the intrinsic nature of the SARS2-S protein.

To further assess differences between the SARS-S and SARS2-S proteins, we next addressed whether these S proteins might differ in their ability to utilize a given level of cell-surface ACE2 or TMPRSS2. Based on lentiviral infection of 293T cells expressing a high and constant level of ACE2 together with a range of expression levels of TMPRSS2 and vice versa (Fig. 1b, c), SARS2-S-mediated infection required higher levels of cell-surface TMPRSS2 expression than SARS-S to attain maximum levels of cell entry. Therefore, it is likely that SARS2-S essentially requires sufficient levels of TMPRSS2 expression.

### SARS2-S D614G variant displays highest entry efficiency among natural S variants.

Next, we investigated using our assay system whether naturally occurring mutations in the S protein affect the cell entry of SARS-CoV-2. We created plasmids expressing five different S variants that were initially identified in China (H49Y[15]), Europe (V367F[16] and D614G[17–19]), and the United States (G476S and V483A[16]) (Fig. 2a), confirmed the levels of expression and virion incorporation of S proteins tested (Supplementary Fig. 3), and examined the effects of these mutations on entry into cells expressing ACE2 and TMPRSS2, by comparison with that of wild-type (WT) S protein. These naturally occurring S mutations resulted in reduced (G476S), equal (V483A), or enhanced (D614G, V367F, and H49Y) cell entry. Remarkably, the D614G mutant displayed the highest level of entry activity among naturally mutated S proteins tested here (~3.5-fold higher than that of the WT protein) (Fig. 2b). Similar difference in cell entry between WT and D614G S-pseudoviruses was clearly observed in human small airway epithelial cells (Supplementary Fig. 4). These results are particularly important because the D614G mutation defines the clade A2a (also called G) that is rapidly spreading worldwide, accounting for the great majority of isolates[20–22].

### D614G mutation confers structural flexibility to S protein.

To analyze differences among WT/mutant SARS2-S and SARS-S, we performed structural analyses on complex models between ACE2 and these S proteins (Fig. 3a, b). Interestingly, the SARS-S trimer showed an open conformation, providing a larger contact area in the receptor-binding domain (RBD) that specifically interacts with ACE2; this might lead to higher accessibility of the SARS-S RBD to ACE2 than that of SARS2-S. This finding is indeed consistent with the recent report that the SARS-CoV-2 S protein has lower ACE2-binding affinity than SARS-CoV S[23]. It is also likely that to a large extent, the structures of SARS2-S mutants reflect differential cell entry (Fig. 3c; see details in the legend). Notably, the aspartic acid residue at position 614 located in the S1 subunit of the WT protein (D614) is able to form a hydrogen bond with a threonine residue at position 859 (T859), as recently reported[24], and/or a salt bridge with a lysine residue at position 854 (K854) located in the S2 subunit of the other protomer (Fig. 3d). This finding suggests in turn that the mutation of this residue to a glycine (G614) can provide flexible space between two protomers due to the short side chain, allowing the S1 subunit to be dissociated more smoothly

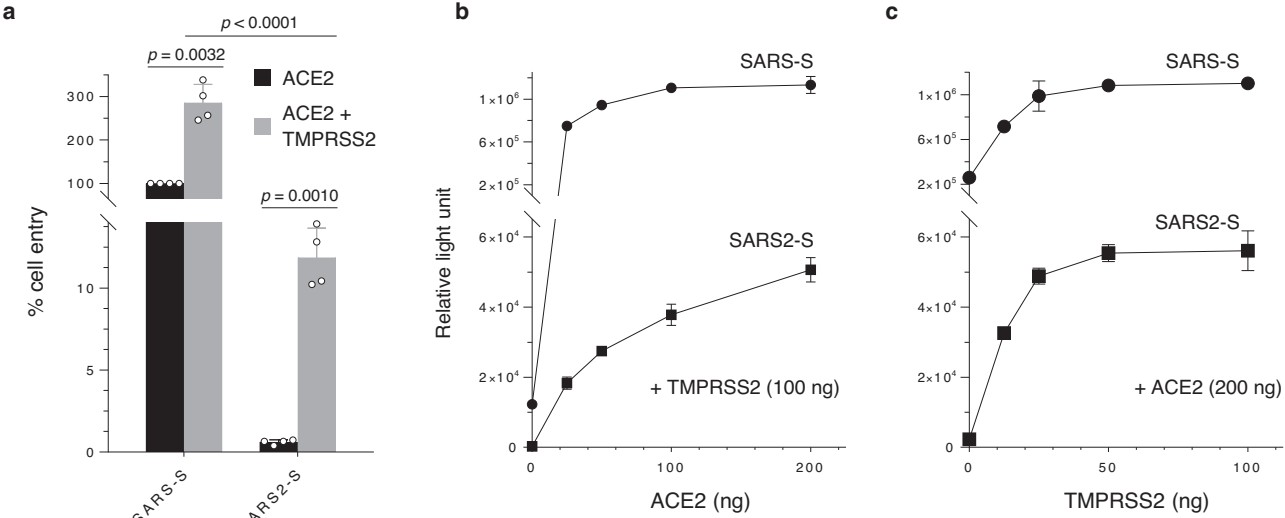

**Fig. 1 SARS2-S-mediated cell entry is highly dependent on TMPRSS2. a** Viruses were prepared by transfection of 293T cells with the HiBiT-tagged lentiviral packaging plasmid, the firefly luciferase-reporter lentiviral transfer plasmid, and either a SARS-CoV S (SARS-S) or SARS-CoV-2 S (SARS2-S) expression plasmid. Viral supernatants were subjected to HiBiT assays, and S-pseudotyped viruses normalized by HiBiT activity were used for infection of 293T cells expressing the host receptor ACE2 alone (black) or coexpressing TMPRSS2 (gray). Cell entry was determined by firefly luciferase activity in cell lysates. Data from four experiments are shown as a percentage of cell entry of SARS-S-pseudotyped viruses into 293T cells expressing ACE2 only (mean ± s.d., $n = 3$ technical replicates). The $p$ value was calculated using a two-tailed paired or unpaired Student's $t$-test. **b**, **c** The effect of ACE2 or TMPRSS2 expression levels on cell entry activity. 293T cells were transfected with a high and constant level of an expression plasmid encoding ACE2 together with increasing levels of a TMPRSS2 expression plasmid (**b**), and vice versa (**c**). Transfected cells were infected with lentiviruses pseudotyped with either SARS-S (circle) or SARS2-S (square), as described in **a**. Data shown are representative of three independent experiments (mean ± s.d., $n = 3$ technical replicates). Source data are provided as a Source Data file.

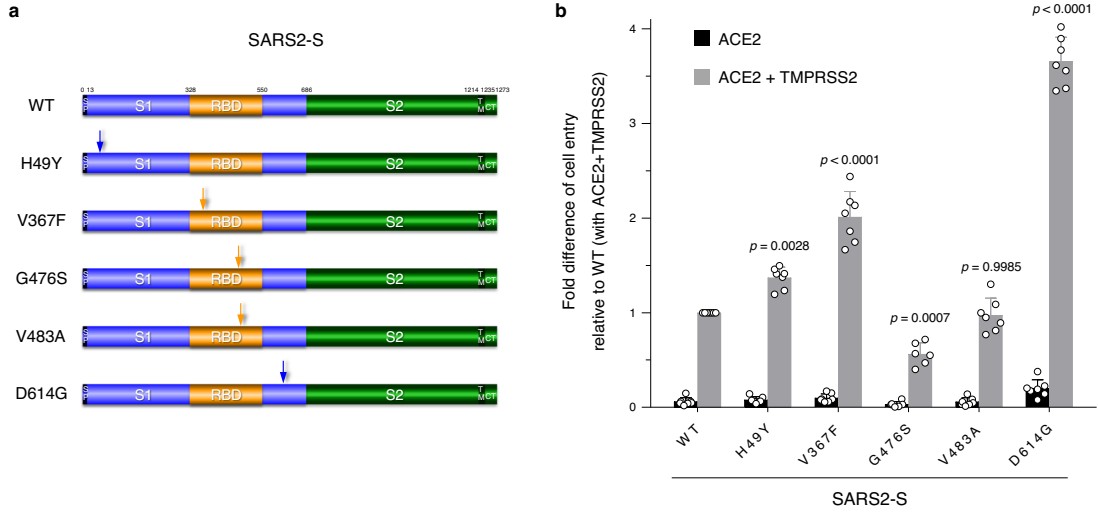

**Fig. 2 SARS2-S D614G variant protein display highest levels of entry activity. a** Schematic illustration of the prototype (wild-type, WT) and globally spread variant SARS2-S proteins. Numbers indicate amino acid positions. The signal peptide (SP), transmembrane domain (TM), cytoplasmic tail (CT), S1 subunit (S1), S2 subunit (S2), and receptor-binding domain (RBD) are indicated. The positions of mutations are indicated by arrows. **b** Functional comparison of the entry activity of WT and mutant SARS2-S proteins. Different S-pseudotyped viruses were prepared as described in **a** and used for infection of 293T cells expressing the host receptor ACE2 alone (black) or coexpressing TMPRSS2 (gray). Cell entry was determined by firefly luciferase activity in cell lysates. Data were shown as the fold difference of cell entry relative to that of WT into 293T cells coexpressing ACE2 and TMPRSS2 (mean ± s.d. from seven independent experiments (except for that of G476S, $n = 6$ with three technical replicates); compared with WT using one-way ANOVA with Dunnett's multiple comparison test. Source data for **b** are provided as a Source Data file.

from the S2 subunit, and/or likely providing conformational flexibility to the overall structure of the S trimer, which might lead to improved affinity for ACE2.

**D614G protein binds ACE2 more efficiently than WT protein.** To test whether D614G mutation could alter binding affinity for

ACE2, we first performed in vitro binding assays using cell lysates expressing ACE2 and SARS2-S proteins. However, our preliminary data did not clearly show the difference in ACE2-binding between WT and D614G in an immunoprecipitation–Western blot assay, probably due to practical limitations of its assay sensitivity. We therefore utilized biolayer interferometry technology

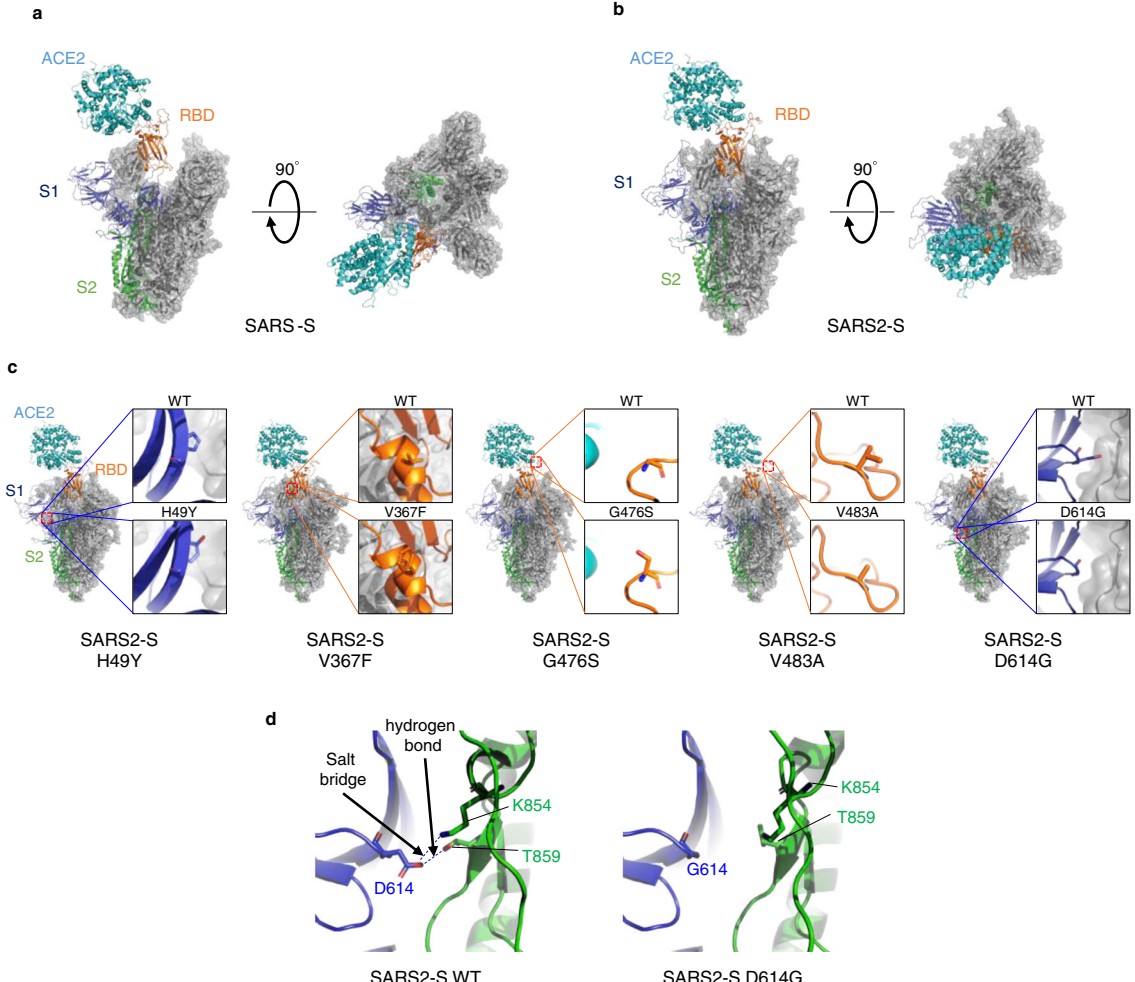

**Fig. 3 SARS2-S D614G protein shows the prominent structural difference. a**, **b** Trimeric structure of the S proteins from SARS-CoV (**a**) and SARS-CoV-2 (**b**) that bind to the host receptor ACE2 (cyan). One protomer of the S protein is shown as a ribbon in navy (S1 subunit), orange (receptor-binding domain (RBD) in S1 subunit), and green (S2 subunit), while two other protomers are shown as gray transparent surfaces and ribbons. The structures are viewed from two different angles. **c** Comparison between wild-type (WT) and mutant S proteins from globally spread SARS-CoV-2 variants. (*Inset*) Enlargement of the region in which each amino acid is mutated, with comparison with the WT S protein. *H49Y*; as the histidine at position 49 is located distant from the RBD and putative cleavage sites, the effect of this mutation on S's function is likely limited. *V367F*; the substitution from a valine to a phenylalanine at position 367 in the RBD introduces a larger side chain at a protomer–protomer interface, which might provide a more rigid RBD structure. *G476S*; the substitution at position 476 in the RBD results in a protruded surface, which appears to interfere with the ACE2–RBD interaction. *V483A*; both valine and alanine residues have short side chains, likely sharing similar phenotypes. *D614G*; details are depicted in Fig. 3d. Note that these structural bases are largely consistent with the results of cell entry activity shown in Fig. 2b. **d** Structural difference between WT and D614G SARS-CoV-2 S proteins. *WT* (left); an aspartic acid (D614) in the S1 subunit (navy) of a protomer binds to a threonine (T859) and/or a lysine (K854) in the S2 subunit (green) of the other protomer though electrostatic interaction between the pairs of these residues. *D614G* (right); the short nonpolar side chain of glycine (G614), which does not bind to T859 and K854, provides flexible space between the two protomers. The figures were drawn with PyMOL ver. 2.4 (https://pymol.org).

to accurately measure the binding affinities of these S proteins to ACE2 (Fig. 4). The Fc-tagged ACE2 dimer was immobilized to anti-Fc biosensors, which were dipped into wells containing different concentrations of a soluble ectodomain of either WT or D614G S protein trimer, and their ACE2-binding affinity was evaluated. The D614G S protein trimer bound to the ACE2 dimer with comparable association kinetics compared with the WT S trimer at all temperatures tested (Fig. 4a–f), but with ~1.8-fold slower dissociation than WT S trimer at both 30 °C (Fig. 4c, d and Supplementary Fig. 5) and 37 °C (Fig. 4e, f), resulting in 2.0 or 1.5-fold increase in binding affinity based on equilibrium dissociation constants values at each temperature. We therefore conclude that the D614G mutation of SARS2-S protein increases binding affinity for the ACE2 receptor, probably resulting from a mutation-induced structural flexibility.

**D614G variant remains susceptible to neutralization by patient sera against prototypic viruses.** Given that the D614G S protein is biologically and structurally different from the WT protein, we hypothesized that this mutation might affect the antigenicity of the S protein. To examine this possibility, we performed neutralization assays to compare the neutralizing sensitivity of the WT and D614G S proteins to anti-SARS-CoV-2 sera. For the neutralization assays, we utilized serum samples derived from five patients confirmed to be infected with prototype viruses and a control serum from a healthy donor. Regardless of serum concentration, the anti-SARS-CoV-2 patient sera but not the control serum efficiently neutralized both viruses pseudotyped with the SARS2-S WT and D614G mutant proteins (Fig. 5). These results indicate that the D614G mutation in the SARS2-S protein maintains neutralization sensitivity to the anti-SARS2-S antibodies, i.e., its antigenicity per se.

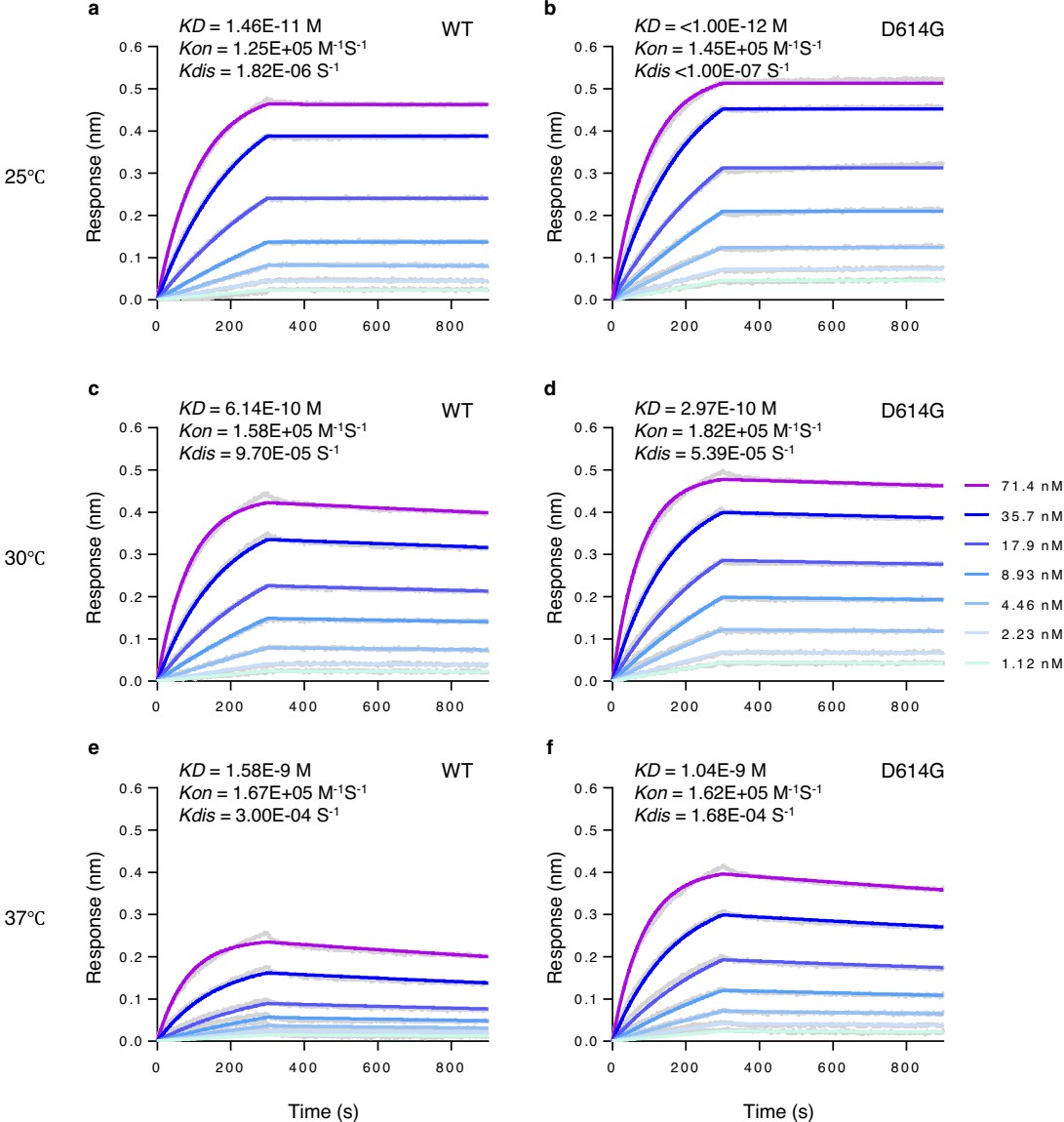

**Fig. 4 SARS2-S D614G protein binds ACE2 with higher affinity than WT S protein.** Binding affinity between dimeric ACE2 and trimeric SARS2-S WT (**a**, **c**, and **e**) or D614G variant (**b**, **d**, and **f**) (two-fold serially diluted from 71.4 nM to 1.12 nM) was evaluated at 25 °C (**a** and **b**), 30 °C (**c** and **d**), or 37 °C (**e** and **f**) using Octet RED96 instrument. The association rate constants ($K_{on}$) and dissociation rate constants ($K_{dis}$) were determined by global fitting of the experimental data using a 1:1 binding model. Equilibrium dissociation constants ($K_D$) were obtained from $K_{dis}/K_{on}$. Data shown are representative sensorgrams from two independent experiments at 25/37°C and from three independent experiments at 30°C using different protein preparations. The difference in $K_D$ and $K_{dis}$ at 30°C was statistically significant ($p = 0.0042$ and $0.0129$, respectively) by a two-tailed unpaired Student's $t$-test, as shown in Supplementary Fig. 5. Source data are provided as a Source Data file.

## Discussion

In this study, we employed a novel entry assay system that we recently developed to quantify cell entry of lentiviral pseudo-viruses[10], in which we can precisely normalize viral input based on a highly accurate standard curve generated by HiBiT luciferase signals in the assays, leading to experimental accuracy in determining levels of cell entry. By using this system, we first showed that SARS2-S-mediated cell entry strongly depends on TMPRSS2 coexpression, without which that of SARS-S still proceeds to some extent. Indeed, SARS-CoV-2 efficiently infects respiratory and intestinal cells[9,25] that coexpress ACE2 and TMPRSS2[8], whereas SARS-CoV can target multiple cell types in several organs[26] expressing ACE2, probably even without TMPRSS2. We also found that SARS2-S-mediated cell entry is considerably lower than that of SARS-S (Fig. 1a and Supplementary Fig. 4),

consistent with the recent report that SARS-CoV replicates ~1000-fold more efficiently than does SARS-CoV-2 in highly permissive human intestinal Caco2 cells[27]. Note that we here tested SARS-S derived from a single representative strain, which could be a limitation of our present findings. Nonetheless, these findings suggest the possibility that SARS2-S is likely less adapted to human cells than SARS-S, and might, at least in part, explain why asymptomatic infection is very high in the case of COVID-19, e.g., in a nursing facility in Washington, USA. (56%; 27 out of 48)[28] and among healthcare workers in the UK (57%; 17 out of 30)[29], whereas it was very rare in the SARS outbreak in Hong Kong in 2003[30].

More importantly, the D614G mutant displayed the highest entry activity among SARS2-S proteins tested in this study (Fig. 2b). Because a single round virus infection is amplified

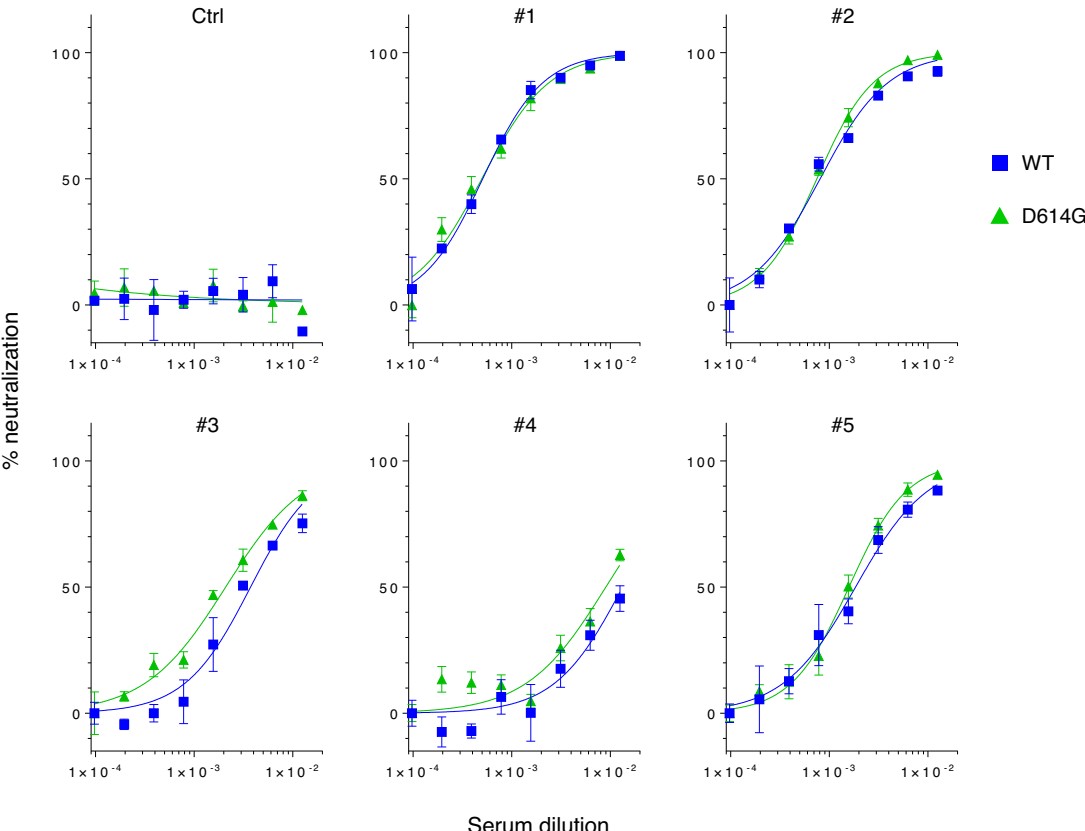

**Fig. 5 SARS-2 S WT and D614G proteins are similarly neutralized by patient sera.** Lentiviruses pseudotyped with either the WT or D614G mutant SARS2-S were preincubated with two-fold serially diluted human sera (80-fold to 10, 240-fold) obtained from a healthy donor (Ctrl) or collected at 15–30 days post-symptom onset from confirmed case patients (#1–#5) infected with the prototypic viruses. The mixture was used for infection of 293T cells coexpressing ACE2 and TMPRSS2, and cell entry levels of pseudoviruses in the presence of diluted patient sera were determined by luciferase assays. Representative data from two independent experiments are shown as percent neutralization (mean ± s.d., $n = 3$ technical replicates). Source data are provided as a Source Data file.

during multiple virus replication cycles, such a seemingly small increase in entry activity can lead to a large difference in viral infectivity in vivo. This finding therefore suggests that the D614G mutant virus might be more transmissible from human to human than the prototype viruses, even though further investigations are necessary to determine the correlation between viral infectivity and transmissibility among humans. In fact, the D614G mutation defines the clade A2a, which has overwhelmingly spread worldwide, accounting for 87% of sequenced cases in New York City (73 out of 84 in one study[22]), 76% of cases in Iceland (438 out of 577, including A2a-derived haplotypes in a study[20]), the vast majority of the recent cases in Japan[24], and almost all of the cases in Italy[24]. It should be noted that in 15 European countries where the A2a clade has been dominant, the estimated doubling time was reported to be ~3 days or less[31], which is significantly shorter than the initial estimation (6.4–7.4 days) in the early outbreak phase in China[2,32].

The aforementioned results indicate that the D614G variant has virological and structural differences from the WT strain. Consistently, our experiments using biolayer interferometry technology demonstrate that the D614G protein trimer has an increased binding affinity for the ACE2 protein dimer in a temperature-dependent manner (Fig. 4). This result indeed reflects the structural prediction that the spatial flexibility of S protein is induced by the D614G mutation, which likely leads to enhanced ACE2-binding (Fig. 3d). In striking contrast, the recent paper published during our revision process, where a surface plasmon resonance technique was used, showed that D614G

rather decreased the affinity for ACE2[33]. This difference is likely due to the different assay systems using different versions of S proteins (with diproline or hexaproline stabilization mutations in their or our study, respectively) so that further investigations are needed to address this important question.

In any case, these findings prompted us to assess whether this mutation might have changed the S protein's antigenicity, as generally represented by reduced antibody-neutralization susceptibility. According to the results of pseudovirus-based neutralization assays, we found that the SARS2-S WT and D614G mutant were similarly susceptible to all confirmed case patient sera raised against the prototypic viruses (Fig. 5). This is particularly important in that the clade-defining D614G mutation will not hinder the current strategy for anti-SARS-CoV-2 drug/vaccine development due to the absence of antigenic alteration in the S protein. It should be noted, however, that all of these infection experiments including cell entry assays were performed by using lentiviral vector-based S-protein-pseudotyped viruses, and therefore the results obtained need to be reproduced by conducting infection experiments using whole SARS-CoV-2 viruses in future studies. Importantly, during and after the submission of our initial preprint, several groups also proved that the D614G mutant virus shows increased infectivity[24,33–36] and retained neutralization sensitivity[33–35], by similarly conducting infectivity assays using S-protein-pseudotyped viruses.

Overall, we speculate that the reason why COVID-19 has become a once-in-a-century pandemic while SARS ended up with epidemics in the past could be, at least in part, that the S protein,

a key viral protein for the first step in transmission, of the causative agent SARS-CoV-2, displays suboptimal levels of cell entry activity as shown in this study. Accordingly, this phenomenon would occasionally result in inefficient replication among humans, presumably leading to a higher rate of asymptomatic infection, as described above. In such cases, unknowingly infected subclinical individuals could be virus spreaders who still have the potential to cause lethal respiratory disease to others, as recently reported[28]. Consequently, during its continuous transmission from human to human, the prototype might have acquired more widely prevalent phenotypes represented by the A2a clade that harbors the D614G mutation, possibly with enhanced speed of global transmission. If this scenario is correct, fatal pathogens for which we should be concerned are those with such characteristics rather than the obviously deadly viruses that can be readily detected. Although in the present case, it is likely that the mutation did not influence the antigenicity of the S protein, we may need further worldwide surveillance for virological changes of SARS-CoV-2 in the human population.

## Methods

**DNA constructs.** The HiBiT-tagged lentiviral packaging plasmid psPAX2-IN/HiBiT, the firefly luciferase-expressing lentiviral transfer plasmid pWPI-Luc2, CMV/R-SARS-S, and the ACE2-expressing plasmid have previously been described elsewhere[10,11,37,38]. The ACE2 expression plasmid pC-ACE2 was created by inserting the Acc65I/XhoI-digested PCR-amplified human ACE2 fragment into the corresponding site of the pCAGGS mammalian expression plasmid[39]. To generate the TMPRSS2 expression plasmid pC-TMPRSS2, total RNA isolated from HepG2 cells was subjected to RT-PCR amplification of the TMPRSS2 gene using specific oligonucleotides, and an amplified fragment digested with BsiWI/XhoI was cloned into Acc65I/XhoI-digested pCAGGS (possible RT-PCR errors or SNPs were corrected by PCR mutagenesis). The SARS-CoV S expression plasmid pC-SARS-S was created by inserting the BsiWI/XhoI-digested PCR-amplified SARS-CoV S fragment of CMV/R-SARS-S into the corresponding site of pCAGGS. The SARS-CoV-2 S expression plasmid pC-SARS2-S was created by inserting the Acc65I/NotI-digested PCR-amplified SARS-CoV-2 S fragment of pCMV3-2019-nCoV-Spike(S1 + S2)-long (Sino Biological; VG40589-UT) into the corresponding site of pCAGGS. The SARS-CoV-2 S mutants (pC-SARS2-S-H49Y, pC-SARS2-S-V367F, pC-SARS2-S-G476S, pC-SARS2-S-V483A, pC-SARS2-S-D614G, or pC-SARS2-S-C1247A), in which positions 49, 367, 476, 483, 614, or 1247 of the S protein were mutated from histidine-to-tyrosine, valine-to-phenylalanine, glycine-to-serine, valine-to-alanine, aspartic-acid-to-glycine, or cysteine-to-alanine, respectively, were created by inserting overlapping PCR fragments into Acc65I/NotI-digested pCAGGS. The plasmid expressing chimeric SARS2-S protein harboring the TM/CT domains of SARS-S, pC-SARS2-S-TM/CT1, was generated by inserting overlapping PCR fragments (amplified from the TM/CT domains of SARS-S and from all other domains of SARS2-S) into Acc65I/NotI-digested pCAGGS. C-terminally T7 epitope (T7e)-tagged S expression plasmids, pC-SARS-S-T7e, pC-SARS2-S-T7e, pC-SARS2-S-C1247A-T7e, and pC-SARS2-S-TM/CT1-T7e were generated by replacing the APOBEC3G gene of pCa-hA3G-T7e[40] with PCR-amplified S fragments. All constructs were verified by a DNA sequencing service (FASMAC). Sequence alignment shown in Supplementary Fig. 2 was performed by Genetyx (v.13.1.2.). A complete list of all primers used in this study is provided in Supplementary Table 1.

**Cell maintenance.** 293T and HepG2 cells, which were originally purchased from the ATCC, were maintained under standard conditions, and routinely tested negative for mycoplasma contamination (PCR Mycoplasma Detection Set, Takara). Human small airway epithelial cells were purchased from ScienCell (#3230) and cultured in poly-L-lysine (ScienCell, #0413)-coated flasks containing Small Airway Epithelial Cell Medium (ScienCell, #3231), 1% Small Airway Epithelial Cell Growth Supplement (ScienCell, #3272), and 1% penicillin/streptomycin solution (ScienCell, #0503).

**Pseudotyped virus preparation and cell entry assays.** To prepare various spike protein-pseudotyped lentiviral luciferase reporter viruses, $1.1 \times 10^5$ 293T cells were cotransfected with 200 ng of spike protein expression plasmids (pC-SARS-S, pC-SARS2-S (-WT, -H49Y, -V367F, -G476S, -V483A, -D614G, or -C1247A), 400 ng of psPAX2-IN/HiBiT, and 400 ng of pWPI-Luc2, using FuGENE 6 (Promega). Sixteen hours later, the cells were washed with phosphate-buffered saline, and 1 ml of fresh complete medium was added. After 24 h, the supernatants were harvested and treated with 37.5 U/ml DNase I (Roche) at 37 °C for 30 min. The lentivirus levels in viral supernatants were measured by the HiBiT assay, as previously described[4]. Briefly, a lentivirus stock with known levels of p24 antigen was serially diluted. Either the standards or viral supernatants containing pseudotyped viruses (25 μl) and LgBiT Protein (1:100)/HiBiT Lytic Substrate (1:50) in Nano-Glo HiBiT Lytic Buffer (25 μl)

(Nano-Glo HiBiT Lytic Detection System; Promega) were mixed and incubated for 10 min at room temperature according to the modified manufacturer's instructions. HiBiT-based luciferase activity in viral supernatants was determined with a Centro LB960 luminometer (Berthold), exported to Microsoft Excel 2016 through MicroWin (v. 4.36.), and translated into p24 antigen levels. To prepare target cells, $2.2 \times 10^5$ 293T cells were cotransfected with increasing amounts of pC-ACE2 and pC-TMPRSS2, together with 700 ng of empty pCAGGS plasmid. After 48 h, transfected cells ($2.2 \times 10^4$) were seeded into 96-well plates, and incubated with lentiviruses pseudotyped with different spike proteins corresponding to 1 ng of p24 antigen. Alternatively, primary human small airway epithelial cells ($2 \times 10^4$) were incubated with increasing p24 amounts of the S-pseudotyped lentiviruses. After 48 h, the cells were lysed in 100 μl of One-Glo Luciferase Assay Reagent (Promega). Firefly luciferase activity was determined with a Centro LB960 luminometer. Note that in this procedure, HiBiT assays using target cell lysates were not employed to determine the levels of cell entry, due to the background caused by non-specific attachment of virions to the cell surface. For immunoblot analysis using virions and producer cells, samples were subjected to gel electrophoresis and transferred to a nitrocellulose membrane. The membranes were probed with an anti-SARS-CoV-2 spike (S2 subunit) mouse monoclonal antibody (1:1,000, GeneTex, GTX632604), an anti-p24 monoclonal antibody (1:1,000), and an anti–β-actin mouse monoclonal antibody (for lysates, 1:5000; Sigma-Aldrich, A5316). Blots were visualized with peroxidase-conjugated AffiniPure goat anti-mouse IgG (1:10000; Jackson ImmunoResearch Laboratories, 115-035-062) and an enhanced chemiluminescence (ECL) western blotting detection system (GE Healthcare), and analyzed using a LAS-3000 imaging system (FujiFilm).

**Spike incorporation assays.** Viral supernatants containing SARS2-S-pseudotyped viruses were prepared as described above, except for using the T7e-tagged version of S expression plasmids. Supernatants were then layered onto 20% (wt/vol) sucrose cushions and subjected to ultracentrifugation ($440,000 \times g$ for 10 min) using an Optima TLX Ultracentrifuge (Beckman Coulter). Pelleted virions were resuspended in SDS-sample buffer and subjected to immunoblot analysis using an anti-T7 epitope mouse monoclonal antibody (1:3000; Novagen, 69522-4) and an anti-p24 monoclonal antibody (1:1,000; Nu24[41]).

**Structural modeling of SARS- and SARS2-S proteins.** The structure-based complex models of ACE2 and the ectodomains (ECs) of SARS- and SARS2-S proteins were built by homology modeling with the Modeller 9v8[42]. The trimeric structure of the EC (PDB code: 6VYB[43]) and that of the RBD-ACE2 complex (PDB: 6M0J[44]) obtained from the Protein Data Bank were used as a template for the modeling. The GenBank reference sequences of SARS2-S and human ACE2 were used for the modeling (GenBank IDs: YP_009724390.1 and NP_001358344.1). The structural models of the H49Y, V367F, G476S, V483A, and D614G mutants were constructed in a similar manner. The complex model of ACE2 and SARS-S's EC was also constructed by using the structures of the EC trimer (PDB: 6CS1[45]), its RBD binding to ACE2 (PDB: 3D0G[46]), and the reference SARS-S sequence (NP_828851.1). The structural figures were generated by using PyMOL ver. 2.4 (https://pymol.org).

**Biolayer interferometry.** The Fc-tagged human ACE2-ectodomain protein dimer (Acrobiosystems, SEC-MALS verified, AC2-H5257) was immobilized to an anti-human IgG Fc capture (AHC) biosensor (FortéBio) for 1 min. ACE2-loaded biosensors were then dipped into wells containing either a His-tagged full-length furin-uncleavable/hexaproline-stabilizing ectodomain of SARS2-S-WT (Acrobiosystems, SEC-MALS verified, SPN-C52H9) or SARS2-S-D614G protein trimer (Acrobiosystems, SEC-MALS verified, SPN-C52H3) at 1.12–71.4 nM in 10X Kinetics Buffer (FortéBio). ACE2-S binding was measured on an Octet RED96e instrument (FortéBio) at 25 °C, 30 °C, or 37 °C with shaking, using a 5-min association step followed by a 15-min dissociation step. The data were corrected by subtracting reference sample, and fit to a 1:1 binding model for determination of affinity constants using Octet Data Analysis Software v11.1 (FortéBio).

**Neutralization assays.** Experiments using human samples were approved by the Medical Research Ethics Committee of the National Institute of Infectious Diseases, Japan. Five serum samples were collected at 15–30 days post-symptom onset from confirmed case patients (obtained in February 2020) who signed an informed consent form, and heat-inactivated at 56 °C for 30 min. Two-fold serially diluted sera were mixed with an equal volume of 1 ng of 24 antigen of the WT or D614G mutant SARS2-S-pseudotyped virus and incubated at 37 °C for 1 h. The mixture was added to 293T cells transiently coexpressing ACE2 and TMPRSS2 (seeded into a 96-well plate). After 48 h, cells were lysed and subjected to luciferase assays, as described above, to determine the levels of neutralization.

**Statistical analyses.** Values are presented as the mean ± s.d. for three to seven independent experiments determined on the basis of pilot experiments to estimate effective numbers. Statistical analyses of the data were performed by using GraphPad Prism version 8.04. Statistical comparisons were made using a two-tailed paired or unpaired Student's t-test or one-way ANOVA with Dunnett's multiple comparison test.

**Reporting summary**. Further information on research design is available in the Nature Research Reporting Summary linked to this article.

## Data availability

Source data are provided with this paper. A complete list of all primers used in this study is provided in Supplementary Table 1. The templates used to create structural models are available from PBD code 6VYB (https://www.rcsb.org/structure/6VYB), 6M0J (https://www.rcsb.org/structure/6M0J), 6CS1 (https://www.rcsb.org/structure/6CS1), and 3D0G (https://www.rcsb.org/structure/3D0G). The GenBank reference sequences of SARS-S, SARS2-S and human ACE2 used for the modeling are available from GenBank IDs: NP_828851.1, YP_009724390.1, and NP_001358344.1, respectively. The codon-optimized SARS2-S sequence is available from Sino Biological's website (https://www.sinobiological.com/cdna-clone/2019-ncov-cov-spike-vg40589-ut). Plasmids are available from K.T. under a material transfer agreement with the National Institute of Infectious Diseases. Source data are provided with this paper.

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

## Acknowledgements

We thank the following physicians who collected blood samples from confirmed case patients with COVID-19: Sakiko Tabata, Mayu Ikeda, and Kaku Tamura (Self-Defense Forces Central Hospital). This work was supported by grants from the Japan Society for the Promotion of Science (KAKENHI, 18K07156) to K.T., Kumamoto University Internal Grant for COVID-19 Research to U.T., the Japan Agency for Medical Research and Development (AMED) Research Program on Emerging and Re-emerging Infectious Diseases (20fk0108293h0001) to Y.I., and AMED Research Program on COVID-19 (19fk0108110 and 20fk0108104) to T.S.

## Author contributions

S.O., Y.Z., H.O., K.S., T.S.T., T.U., Y.I., and K.T. performed experiments and analyzed the data. H.O., S.K., T.U., Y.I., T.S., and K.T. interpreted and discussed the data. K.I. and K.M. provided reagents. K.T. conceived the study, supervised the work, and wrote the paper. All authors read and approved the final manuscript.

## Competing interests

The authors declare no competing interests.
