## [Peer Review File · Nature Communications]

Reviewers' comments:

Reviewer #1 (Remarks to the Author):

In the manuscript entitled: "Naturally mutated spike proteins of SARS-CoV-2 variants show differential levels of cell entry" Ozono et al., employ a novel measurement system to quantify S-mediated cell entry which was recently established for the same laboratory and, as far as this reviewer could tell, it is not yet peer-reviewed published. This system allows to precisely normalize input virus doses which is critical to draw meaningful conclusions in comparative studies like this manuscript. Despite the experiments are well done this reviewer thinks the results are not novel enough in the current version of the manuscript.

Major Points

Fig. 1a-c, Fig. S1

The authors examined the viral entry of lentiviruses pseudotyped with SARS-CoV S or SARS-CoV-2 S and also tested whether the co-expression of ACE2 and TMPRSS2 could mediate efficient infection of 293T cells with SARS2-S-pseudotyped lentiviruses.

The results show that SARS2 enters cells much less efficiently than SARS and the dual expression of ACE2 and TMPRSS2 improve SARS2 entry but still is much lower than SARS.

These are not novel results, they were already published by Hoffmann et al., Cell, 2020. In the present manuscript it would be helpful to have a comparison between 293T cells and lung cells using this new method that allow precise normalization of virus inputs.

Fig. 1e.

Shows differently entry levels for naturally occurs mutations.

It is already published that mutation D614G increases infectivity (Korber et al., Cell, 2020) which is the only mutant further investigated in the manuscript because of its interest phenotype.

Fig S2

This reviewer believes that a comparison with SARS in Fig S2C is missing to conclude that low rate cell entry is not due to differences in S incorporation into virions.

Do the authors see same differences in entry when comparing 293T and lung cells or cells from the airway epithelial?

Fig 2

Structural analyses on complex models between ACE2 and naturally occur S mutants. The authors hypothesized that the mutation D614G can provide flexible space between two protomers due to the short G side chain which might lead to improved accessibility of ACE2 into the RBD.

To test this possibility, they performed *in vitro* binding and did not observe any difference in ACE2 binding between WT and D614G (Extended Data Fig. 4).

If the amount of proteins use in the *in vitro* assay was above the Kd then is expected to see no differences. This reviewer thinks that the assay to do is to compare affinities between D614 and G 614 for ACE2 using octet or biacore.

The sentence "Nevertheless, the aforementioned structural data are fully consistent with the high entry activity of the virus harboring this mutation (Fig. 1e)" should be remove since from the structural data is coming a hypothesis that actually the authors couldn't demonstrate.

Fig 3.

The authors hypothesized that mutation D614G might affect the antigenicity of the S protein since pseudo-viruses carrying this variant are biologically and structurally different from wild-type. The authors performed neutralization assays to compare the neutralizing sensitivity of the WT and D614G S proteins to anti-SARS-CoV-2 sera.

Do the authors know the status of D614G in the individuals from where the sera used in the experiments is coming from? In other words, if the individuals were infected with a mixed population of wt and mutant virus then is expected that the sera will contained antibodies that neutralize both pseudo-types equally and the question about the different antigenicity is therefore not conclusively answered.

Minor Points

-Fig S2

Is missing the key colors for the residues labeled.

Reviewer #2 (Remarks to the Author):

In this short communication, Ozono and co-authors compared the infectivity of lentiviruses pseudotyped with SARS-CoV and SARS-CoV-2 spike (S) protein (SARS and SARS2, respectively) and examined the effect of naturally occurring mutations in SARS2 S on virus entry measured by HiBiT tag-based single cycle infectivity assay. This comparison revealed much more efficient SARS S-mediated cell entry compared to SARS2 S and a marked dependence of the latter protein's function of TMPRSS2 expression, leading the authors to conclude that SARS2 is less adapted to infect human cells than SARS. Among the naturally occurring SARS2 spike mutations, the D614G mutation markedly enhanced pseudovirus infectivity, albeit not nearly reaching the level observed for SARS. Structural modeling predicted that this mutation increases structural flexibility of S trimer and thereby facilitates ACE2 binding, however, such an effect was not experimentally observed. Finally, the authors examined the susceptibility of WT and D614G SARS2 S to neutralizing sera and found that this mutation does not confer neutralization resistance in a single cycle infectivity assay.

Overall, this study is timely and significant, as it quantitatively examines the relative entry efficiency of SARS2 and naturally occurring S mutants and their neutralization by convalescent sera. The finding of this study should be of interest to virologists working on COVID-19 and for a broader audience interested in this virus and coronaviruses in general. The paper is well written.

It is difficult to fully evaluate the novelty of this study, as there is a recent Cell publication (<https://doi.org/10.1016/j.cell.2020.06.043>) and several papers deposited to BioRxiv on the same subject, but I trust the editor to make this determination. I have the following major concerns related to the experiments and their interpretation.

1. Comparison of one SARS and SARS-2 strain each does not allow for generalization of the experimental findings. If no other strains can be compared, I would at least try to soften the conclusion that SARS S mediates more efficient cellular entry than SARS2 S.
2. The extents of S protein incorporation for SARS and SARS2 are not properly assessed in Extended Figure 2c. There is no SARS S control to judge the relative incorporation of WT SARS2 S protein or its mutants. This important control should be included to meaningfully compare the respective infectivity results.
3. Similarly, Figure 1e examines the effect of several mutations in SARS2 S protein on cellular entry (=infectivity) without assessing their levels of expression and incorporation into pseudoviruses.

Reviewer #3 (Remarks to the Author):

NCOMMS-20-26731-T

This submission reports several findings pertaining to SARS-CoV-2 entry. One is that SARS-CoV-2 pseudovirus entry is lower than SARS-CoV-1, and that TMPRSS2 facilitates SARS-CoV-2 entry more than SARS-CoV-1, at least in a narrow 293T cell context. Another is that a common isolate of SARS-CoV-2 that has a G at position 614 (G614) has more transducing activity than the D614 form. A third finding is that the D614 and G614 forms are similarly neutralized by patient convalescent sera.

Comments:

1. A central comment concerns the originality of the findings. The authors do not cite bioRxiv

2020.06.12.148726; doi: <https://doi.org/10.1101/2020.06.12.148726> or bioRxiv 2020.06.14.151357; doi: <https://doi.org/10.1101/2020.06.14.151357>, both of which document the infection enhancing effect of the D614G. The authors also do not cite bioRxiv 2020.07.04.187757; doi: <https://doi.org/10.1101/2020.07.04.187757> which contain the information that is provided in figures 2 and 3 of this submission, notably the similar susceptibility of D614 and G614 to neutralization by antibodies. In sum, the majority of the findings reported here have been reported previously. There could be debates about biorxiv preprints being preliminary and therefore not relevant to questions about originality of findings, but the authors do cite several other less relevant bioRxiv and medRxiv preprints in their reference list, making it puzzling that the principal meaningful ones are not present.

2. A second comment concerns whether the work supports the conclusions and claims made in the report. Here there are several specific notes.

- a. The authors emphasize a novel measurement for viral entry. The novelty is that HIV core proteins (integrase) can be measured using a luminescence assay. It should be noted how this novelty advances understanding of viral entry, beyond that supplied by conventional measurements such as p24 ELISAs. At present, it is not clear how the innovation of HiBiT brings out new theoretical insights.
- b. The authors claim that their novel measurements provide enhanced experimental accuracy (line 13, page 3 and elsewhere). What is the evidence to support this claim?
- c. There are claims that the experiments involve normalized virus input doses, but this does not appear in the report, and importantly, the ratios of HiBiT to S proteins are not provided for normalization purposes.
- d. It is claimed on lines 10-12, page 4, that a hypothesis about SARS-2 vs. SARS-1 spike incorporation is to be tested. This hypothesis, however, is not tested. Instead, the results show that C1247A and TM/CT1 spikes do not affect SARS-2 S2 levels in particles. The rationales for evaluating these altered spikes is entirely unclear and has no apparent bearing on SARS-1 spikes incorporation into pseudoparticles. The conclusions on lines 18-20, page 4 are not supported.
- e. It is claimed that SARS-2 spike mediated infection required higher levels of ACE2 and TMPRSS2 than SARS-1 spikes (lines 3-4, page 5). This claim is not supported by data. Statistics and biological repeats are needed because inspection of Fig. 1 data do not bring much confidence for this claim.
- f. The nearly exclusive utilization of 293T cells limits the findings and it is unclear whether many claims can be generalized beyond the 293T cell culture condition.

3. A third comment concerns data analysis and presentation. Several graphs in Figs. 1 and supplementary Figs. 1 and 2 present normalized data, and it is not clear how "zero percent" is established. The same applies to supplementary Fig. 3, where the negative control condition is not made clear. Typically, in pseudovirus transduction assays, "zero" is established using pseudoviruses that lack viral surface glycoproteins ("bald" particles). Was this done? This is important, because claims are made about the differences between SARS-1 and SARS-2 pseudovirus entry, and they have to be considered in conjunction with appropriate background value determinations.

4. Additional concerns include supplementary Fig. 2, which lacks rationale. It is not clear why TM and cyto tail amino acid substitutions are relevant to this study. Also, supplementary Fig. 4 data provide a good start to evaluating ACE2-spike affinities but the co-IP approaches are not quantitative and cannot be used to assess relative strengths of ACE2 binding. More sophisticated quantitative measures are required.

First, we would like to sincerely thank all the reviewers for the constructive and critical comments, despite this hard time of COVID-19 pandemic. Details of our response to the referees' concerns are as follows:

Reviewer #1 (Remarks to the Author)

In the manuscript entitled: “Naturally mutated spike proteins of SARS-CoV-2 variants show differential levels of cell entry” Ozono et al., employ a novel measurement system to quantify S-mediated cell entry which was recently established for the same laboratory and, as far as this reviewer could tell, it is not yet peer-reviewed published. This system allows to precisely normalize input virus doses which is critical to draw meaningful conclusions in comparative studies like this manuscript. Despite the experiments are well done this reviewer thinks the results are not novel enough in the current version of the manuscript.

We thank the reviewer’s comment. Our measurement system has been recently published in *J Biol Chem*, doi:10.1074/jbc.RA120.013887. The novelty of our results is explained in the comment#2.

Major Points

1. Fig. 1a-c, Fig. S1

The authors examined the viral entry of lentiviruses pseudotyped with SARS-CoV S or SARSCoV- 2 S and also tested whether the co-expression of ACE2 and TMPRSS2 could mediate efficient infection of 293T cells with SARS2-S-pseudotyped lentiviruses. The results show that SARS2 enters cells much less efficiently than SARS and the dual expression of ACE2 and TMPRSS2 improve SARS2 entry but still is much lower than SARS.

These are not novel results, they were already published by Hoffmann et al., Cell, 2020. In the present manuscript it would be helpful to have a comparison between 293T cells and lung cells using this new method that allow precise normalization of virus inputs.

The Reviewer’s comments on Fig. 1a-c “These are not novel results, they were already published by Hoffmann et al., Cell, 2020” is incorrect. I have no idea why the Reviewer misunderstood like this. Hoffmann’s paper rather showed no difference between SARS-S and SARS2-S in entry (see the right figure). Indeed, they did not normalize input pseudoviruses in their entry assays, in which they used replication-defective VSV particles bearing S proteins so that, unlike lentivirus particles, the sole method to normalize VSV-based pseudoviruses is a real-time RT-PCR that was not performed in their paper. Therefore, our finding that SARS2-S-mediated cell entry is much less efficient than that of SARS-S by using our system is one of the important novelties in our paper, which is discussed in the last

[Redacted]

paragraph of the Discussion. With respect to an infection experiment using primary cells, we have added the SARS-S data obtained in the same set of infection experiment using human airway cells in Extended Data Fig. 4.

2. Fig. 1e.

Shows differently entry levels for naturally occurs mutations.

It is already published that mutation D614G increases infectivity (Korber et al., Cell, 2020) which is the only mutant further investigated in the manuscript because of its interest phenotype.

We agree that Korber's Cell paper showing increased D614G infectivity was already published on July 3 online; however, the bioRxiv version of our paper was published on June 15. <https://www.biorxiv.org/content/10.1101/2020.06.15.151779v1> In fact, the Cell's preview (entitled "Making Sense of Mutation: What D614G Means for the COVID-19 Pandemic Remains Unclear" <https://doi.org/10.1016/j.cell.2020.06.040>) for Korber's paper indeed cited our bioRxiv preprint. Still, unlike the Cell paper plus other bioRxiv papers that were posted with the same timing as ours, we did not focus only on the comparison between the prototype and D614G mutant of SARS-CoV-2. As described above in the response to Reviewer #1's comments#1, this present study is the first report to show the critical difference between SARS-S and SARS2-S in cell entry efficiency. Besides, to our knowledge, this is also the first description of a much stronger dependency of SARS2-S-mediated entry on TMPRSS2 expression than that of SARS-S-mediated entry.

3. Fig S2

This reviewer believes that a comparison with SARS in Fig S2C is missing to conclude that low rate cell entry is not due to differences in S incorporation into virions.

Do the authors see same differences in entry when comparing 293T and lung cells or cells from the airway epithelial?

We appreciate the Reviewer's comment. This comparison is indeed our dilemma for the following reasons; The anti-S2 subunit monoclonal antibody (GeneTex, GTX632604) that we used in this study was originally generated against SARS-S, but not SARS2-S, so that the antibody can naturally react more efficiently with SARS-S than with SARS2-S. Even by using its tagged version using an anti-tag antibody, the SARS2-S incorporated into virions is completely cleaved by host cell proteases, whereas that of SARS-S is uncleaved and results in a two-times larger band size in immunoblot analyses (as reported by Hoffman et al. (Cell, 2020)), making it difficult to quantitatively compare these two different spike proteins. Instead of comparing them, we had chosen to compare the virion incorporation of SARS2-S WT and chimeric mutants that harbor SARS-S type of cytoplasmic tail (CT) or CT/transmembrane (TM), because accumulated evidence has shown that the compatibility of a variety of viral envelopes with lentiviral virions is determined by their CT domains. To emphasize this, we have added the following sentence with a couple of citations; "Nevertheless, ~25-fold differences in cell entry between SARS-S and SARS2-S pseudoviruses were observed (Fig. 1a),

leading to the hypothesis that SARS2-S might be less compatible with lentiviral particles. Because a large body of literature has shown that lentiviral compatibility of a variety of viral envelopes is determined by their cytoplasmic tail (CT) domains¹²⁻¹⁴, we compared the sequences of these domains in S proteins and found that...” in the text (page 4, lines 9-15). Nevertheless, based on the fact that we did not directly and quantitatively compare the virion incorporation of these two spike proteins, we decided to change the wording of conclusions as follows (page 4, line 20 – page 5, line 1); “These results suggest that the significantly lower rate of cell entry of the SARS2-S pseudovirus was not due to the incompatibility between SARS2-S and a lentiviral vector but rather to the intrinsic nature of the SARS2-S protein.” Also, we added the following sentence “Note that SARS2-S incorporated into virions is completely cleaved by host cell proteases, whereas that of SARS-S is uncleaved and results in a two-times larger band size in immunoblot analyses (as reported by Hoffman et al. (Cell, 2020)), making it unable to directly compare these S proteins.” in the legend of Extended Data Fig. 2.

As for the infection experiment using human airway cells, we have added the data in Extended Data Fig. 4., as described in the response to the comments#1

4. Fig 2

Structural analyses on complex models between ACE2 and naturally occur S mutants. The authors hypothesized that the mutation D614G can provide flexible space between two protomers due to the short G side chain which might lead to improved accessibility of ACE2 into the RBD. To test this possibility, they performed in vitro binding and did not observe any difference in ACE2 binding between WT and D614G (Extended Data Fig. 4). If the amount of proteins use in the in vitro assay was above the Kd then is expected to see no differences. This reviewer thinks that the assay to do is to compare affinities between D614 and G 614 for ACE2 using octet or biacore.

The sentence “Nevertheless, the aforementioned structural data are fully consistent with the high entry activity of the virus harboring this mutation (Fig. 1e)” should be remove since from the structural data is coming a hypothesis that actually the authors couldn’t demonstrate.

“The assay to do is to compare affinities between D614 and G614 for ACE2 using octet or biacore” is unlikely for the following reasons: As reported by Shang *et al.* (PNAS, 117:11727-11734, 2020), their results obtained in IP-based comparative analysis of receptor binding affinity of SARS-S and SARS2-S proteins contradicted their own recent results obtained from Biacore based on surface plasmon resonance (SPR) by using purified RBD and ACE2 proteins (Nature, 581:221–224, 2020), and they therefore concluded that “whereas SARS-CoV-2 RBD has higher ACE2 binding affinity than SARS-CoV RBD, SARS2-S has lower ACE2 binding affinity than SARS-S”. This contradiction is somewhat predictable because the interaction of purified RBD-ACE2 proteins they observed in Nature paper was an RBD monomer-vs-ACE2 dimer interaction that is different from their native binding conformation.

For example, in the case of soluble versions of HIV-1 Envelope glycoprotein (Env), it consists of full-

length gp120 and most of the gp41 ectodomain in which the cytoplasmic tail and transmembrane regions of gp41 are deleted. The natural association between gp120 and gp41 in the functional Env spike is non-covalent and labile, thereby recombinant expression of cleaved Env results in dissociation of the hetero-dimeric subunits (Sharma *et al.* Cell Rep. 11:539-550, 2015). Exactly the same thing can be said for S1 and S2 proteins of SARS-CoV-2 (which correspond to HIV-1 gp120 and gp41, respectively).

For this reason, even a commercially available SARS2-S protein, such as a purified SARS2-S trimer (Acro Biosystems, SPN-C52H8), as well as the similar SARS2-S trimer just published by Yurkovetskiy *et al.* (Cell, <https://doi.org/10.1016/j.cell.2020.09.032> Sep. 15, 2020), has mutations in its cleavage site, resulting in non-cleaved precursor of SARS2-S that is stable but structurally different from the native form of this protein, which will make it difficult to precisely compare the ACE2 binding affinity of SARS-S and SARS2-S proteins, even by performing SPR-based Biacore.

Therefore, it would be technically difficult to reproduce the actual native binding between ACE2 dimer and either WT or D614G trimer by using purified proteins through either Biacore or Octet.

As suggested, the sentence “Nevertheless, the aforementioned structural data are fully consistent with the high entry activity of the virus harboring this mutation (Fig. 1e)” was accordingly removed.

5. Fig 3.

The authors hypothesized that mutation D614G might affect the antigenicity of the S protein since pseudo-viruses carrying this variant are biologically and structurally different from wildtype. The authors performed neutralization assays to compare the neutralizing sensitivity of the WT and D614G S proteins to anti-SARS-CoV-2 sera.

Do the authors know the status of D614G in the individuals from where the sera used in the experiments is coming from? In other words, if the individuals were infected with a mixed population of wt and mutant virus then is expected that the sera will contained antibodies that neutralize both pseudo-types equally and the question about the different antigenicity is therefore not conclusively answered.

The individuals from which the sera were derived were not infected with a mixed population of wt and mutant virus. As described in the text, the sera we used is those derived from patients infected with the Wuhan (WT) prototypic virus in Japan in early (or before) February 2020 when D614G mutant viruses did not even exist in the world.

Minor Points

-Fig S2

Is missing the key colors for the residues labeled.

We have changed the default color setting, and added a new color scheme into the figure legend as follows; “polar neutral in green, aliphatic/hydrophobic in orange, aromatic/hydrophobic in purple, basic in red, and acidic in blue.”

Reviewer #2 (Remarks to the Author)

In this short communication, Ozono and co-authors compared the infectivity of lentiviruses pseudotyped with SARS-CoV and SARS-CoV-2 spike (S) protein (SARS and SARS2, respectively) and examined the effect of naturally occurring mutations in SARS2 S on virus entry measured by HiBiT tag-based single cycle infectivity assay. This comparison revealed much more efficient SARS S-mediated cell entry compared to SARS2 S and a marked dependence of the latter protein's function of TMPRSS2 expression, leading the authors to conclude that SARS2 is less adapted to infect human cells than SARS. Among the naturally occurring SARS2 spike mutations, the D614G mutation markedly enhanced pseudovirus infectivity, albeit not nearly reaching the level observed for SARS. Structural modeling predicted that this mutation increases structural flexibility of S trimer and thereby facilitates ACE2 binding, however, such an effect was not experimentally observed. Finally, the authors examined the susceptibility of WT and D614G SARS2 S to neutralizing sera and found that this mutation does not confer neutralization resistance in a single cycle infectivity assay.

Overall, this study is timely and significant, as it quantitatively examines the relative entry efficiency of SARS2 and naturally occurring S mutants and their neutralization by convalescent sera. The finding of this study should be of interest to virologists working on COVID-19 and for a broader audience interested in this virus and coronaviruses in general. The paper is well written.

It is difficult to fully evaluate the novelty of this study, as there is a recent Cell publication (<https://doi.org/10.1016/j.cell.2020.06.043>) and several papers deposited to BioRxiv on the same subject, but I trust the editor to make this determination. I have the following major concerns related to the experiments and their interpretation.

We thank the Reviewer for the positive evaluation of our work, and for pointing to shortcomings in the initial version of the manuscript.

1. Comparison of one SARS and SARS-2 strain each does not allow for generalization of the experimental findings. If no other strains can be compared, I would at least try to soften the conclusion that SARS S mediates more efficient cellular entry than SARS2 S.

We are grateful for the Reviewer's suggestion. We chose the best-characterized SARS-CoV representative strain Urbani; however, we indeed agree with the Reviewer and have added the related sentence in page 7 lines 21-22, as follows: “Note that we here tested SARS-S derived from a single representative strain, which could be a limitation of our present findings.”

2. The extents of S protein incorporation for SARS and SARS2 are not properly assessed in Extended Figure 2c. There is no SARS S control to judge the relative incorporation of WT SARS2 S protein or its mutants. This important control should be included to meaningfully compare the respective infectivity results.

We appreciate the Reviewer's comment, which was answered in the response to the Reviewer#1's comment#3

3. Similarly, Figure 1e examines the effect of several mutations in SARS2 S protein on cellular entry (=infectivity) without assessing their levels of expression and incorporation into pseudoviruses.

We thank the Reviewer for the comment. We have performed the experiments and added the data in new Extended Data Fig. 3.

Reviewer #3 (Remarks to the Author)

NCOMMS-20-26731-T

This submission reports several findings pertaining to SARS-CoV-2 entry. One is that SARS-CoV-2 pseudovirus entry is lower than SARS-CoV-1, and that TMPRSS2 facilitates SARS-CoV-2 entry more than SARS-CoV-1, at least in a narrow 293T cell context. Another is that a common isolate of SARS-CoV-2 that has a G at position 614 (G614) has more transducing activity than the D614 form. A third finding is that the D614 and G614 forms are similarly neutralized by patient convalescent sera.

Comments:

1. A central comment concerns the originality of the findings. The authors do not cite bioRxiv 2020.06.12.148726; doi: <https://doi.org/10.1101/2020.06.12.148726> or bioRxiv 2020.06.14.151357; doi: <https://doi.org/10.1101/2020.06.14.151357>, both of which document the infection enhancing effect of the D614G. The authors also do not cite bioRxiv 2020.07.04.187757; doi: <https://doi.org/10.1101/2020.07.04.187757> which contain the information that is provided in figures 2 and 3 of this submission, notably the similar susceptibility of D614 and G614 to neutralization by antibodies. In sum, the majority of the findings reported here have been reported previously. There could be debates about biorxiv preprints being preliminary and therefore not relevant to questions about originality of findings, but the authors do cite several other less relevant bioRxiv and medRxiv preprints in their reference list, making it puzzling that the principal meaningful ones are not present.

We are just wondering whether or not the Reviewer#3 noticed the bioRxiv version of our paper.

<https://www.biorxiv.org/content/10.1101/2020.06.15.151779v1> In fact, we posted this manuscript on bioRxiv, exactly at the same timing (June 15) as the other groups did (bioRxiv 2020.06.12.148726; doi: <https://doi.org/10.1101/2020.06.12.148726> or bioRxiv 2020.06.14.151357; doi: <https://doi.org/10.1101/2020.06.14.151357>). Then, after being rejected by Nature Medicine to which we initially submitted the paper, we transferred the manuscript to Nature Communications (July 3) before the paper that the Reviewer#3 suggested us to cite was posted on bioRxiv 2020.07.04.187757; doi: <https://doi.org/10.1101/2020.07.04.187757> . Accordingly, the reviewer mentioned that we cited several other “less relevant” bioRxiv and medRxiv preprints in our reference list, those were the relevant citations at the timing of the submission of this manuscript. In the revised manuscript, we have newly included the related papers; “Importantly, after the submission of our preprint, several groups also proved that the D614G mutant virus shows increased infectivity³³⁻³⁷ and retained neutralization sensitivity³³⁻³⁵, by similarly conducting infectivity assays using S-protein-pseudotyped viruses.”

2. A second comment concerns whether the work supports the conclusions and claims made in the report. Here there are several specific notes.

a. The authors emphasize a novel measurement for viral entry. The novelty is that HIV core proteins (integrase) can be measured using a luminescence assay. It should be noted how this novelty advances understanding of viral entry, beyond that supplied by conventional measurements such as p24 ELISAs. At present, it is not clear how the innovation of HiBiT brings out new theoretical insights.

We thank the Reviewer for the comment. As anybody who has ever performed p24 ELISAs may notice its intrinsic limitation, the standard curve in the assay is not always linearly generated, unlike that shown in the instruction manual of commercially available kits, and this is mainly due to multiple washing steps. Besides, because of a very narrow range of detectable concentrations, the assay necessarily requires extensive dilutions of samples, often causing small technical errors that can be easily accumulated during sample preparation. In contrast, our HIV/lentivirus-HiBiT detection system that we have recently established (Ozono *et al.*, JBC. 295, 13023-13030 (2020)) shows a highly stable accuracy in the standard curve ($R^2 = 0.999-1.0$) with a wide range of detectable HiBiT activity (linear scale over 6 orders of magnitude; see below), enabling us to skip sample dilution and thereby allowing one-step procedure. We have newly added the related sentence in page 7 lines 11-12; “based on a highly accurate standard curve generated by HiBiT signals in the assays”

[Redacted]

b. The authors claim that their novel measurements provide enhanced experimental accuracy (line 13, page 3 and elsewhere). What is the evidence to support this claim?

As described in the above response to #2a, our HiBiT lentiviral system allowed us to precisely normalize input virus doses. This is particularly important when two or more different viruses are compared for infection and when the difference of the resultant infectivity is as small as that observed in this study (~3.5 fold).

c. There are claims that the experiments involve normalized virus input doses, but this does not appear in the report, and importantly, the ratios of HiBiT to S proteins are not provided for normalization purposes.

We indeed described the normalized virus input doses for infection (corresponding to 1 ng of p24 antigen in 293T cells) in page 3 in the Methods. In order to clarify this, we have added the following words (underlined) in the Methods (page 3. lines 4-5); “HiBiT-based luciferase activity in viral supernatants was determined with a Centro LB960 luminometer (Berthold) and translated into p24 antigen levels.” As for the ratios of HiBiT to S proteins, we did not consider them because we used a fixed recipe for cotransfection, and as long as we use the same amounts of virus input doses, almost the same relative light units were reproducibly generated by infection.

d. It is claimed on lines 10-12, page 4, that a hypothesis about SARS-2 vs. SARS-1 spike incorporation is to be tested. This hypothesis, however, is not tested. Instead, the results show that C1247A and TM/CT1 spikes do not affect SARS-2 S2 levels in particles. The rationales for evaluating these altered spikes is entirely unclear and has no apparent bearing on SARS-1 spikes incorporation into pseudoparticles. The conclusions on lines 18-20, page 4 are not supported.

As described in a response to the Reviewer#1's comment#3, a large body of evidence showed that the virion incorporation of a variety of viral envelopes are determined by their cytoplasmic tail (CT) and/or transmembrane (TM) domains, so that we created the chimeric construct of S proteins.

e. It is claimed that SARS-2 spike mediated infection required higher levels of ACE2 and TMPRSS2 than SARS-1 spikes (lines 3-4, page 5). This claim is not supported by data. Statistics and biological repeats are needed because inspection of Fig. 1 data do not bring much confidence for this claim.

The requirement of TMPRSS2 for SARS2-S-mediated entry is obvious in Fig. 1a with statistical significance, which was strongly confirmed in Fig. 1c in a dose dependent manner. As for ACE2, we decided not to emphasize the requirement of its higher levels so that we have changed the related sentence in page 5, lines 7-10, as follows; "SARS2-S-mediated infection required higher levels of cell-surface TMPRSS2 expression than SARS-S to attain maximum levels of infectivity. Therefore, it is likely that SARS2-S essentially requires sufficient levels of TMPRSS2 expression."

f. The nearly exclusive utilization of 293T cells limits the findings and it is unclear whether many claims can be generalized beyond the 293T cell culture condition.

This is exactly why we conducted infection experiments using human small airway epithelial cells in Extended Data Fig. 4.

3. A third comment concerns data analysis and presentation. Several graphs in Figs. 1 and supplementary Figs. 1 and 2 present normalized data, and it is not clear how "zero percent" is established. The same applies to supplementary Fig. 3, where the negative control condition is not made clear. Typically, in pseudovirus transduction assays, "zero" is established using pseudoviruses that lack viral surface glycoproteins ("bald" particles). Was this done? This is important, because claims are made about the differences between SARS-1 and SARS-2 pseudovirus entry, and they have to be considered in conjunction with appropriate background value determinations.

We have never established such a "zero-percent" definition, because e.g., in Fig.1 and Extended Data Fig. 2 using 293T cells, the actual relative light unit (RLU) values of firefly luciferase from SARS2-S pseudoviruses are ~50,000 (as shown in Fig. 1b), whereas those from spikeless (bald) viruses, which are completely equivalent to those of uninfected cells, are always less than 100 RLU in our experimental conditions. Therefore, when displayed as a percentage, the control value becomes less than 0.2%, looking like zero in bar graphs if the value is plotted in linear scale. This normally happens in infection with luciferase-reporter lentiviruses when sufficiently high levels of RLU are observed. With regard to Extended Data Fig. 1 using human primary airway cells, it does not present normalized data, but shows the actual measured values so that we have added the following words (underlined) into the figure legend; "The dashed line indicates the negative control background generated by spikeless lentiviruses."

4. Additional concerns include supplementary Fig. 2, which lacks rationale. It is not clear why TM and cyto tail amino acid substitutions are relevant to this study. Also, supplementary Fig. 4 data provide a good start to evaluating ACE2-spike affinities but the co-IP approaches are not quantitative and cannot be used to assess relative strengths of ACE2 binding. More sophisticated quantitative measures are required.

The first part is the same with the question#2d, which was answered above. As for Extended Data Fig. 4 (new Extended Data Fig. 5), this is also answered in the response to the Reviewer#1's comment#4.

We thank all the reviewers for their insightful comments and suggestions for improvement that have greatly improved our paper. We hope that our manuscript will now be acceptable for publication in its current form.

REVIEWER COMMENTS

Reviewer #1 (Remarks to the Author):

Comments

2. Fig. 1e. Shows differently entry levels for naturally occurs mutations.

It is already published that mutation D614G increases infectivity (Korber et al., Cell, 2020) which is the only mutant further investigated in the manuscript because of its interest phenotype.

We agree that Korber's Cell paper showing increased D614G infectivity was already published on July 3 online; however, the bioRxiv version of our paper was published on June 15.

<https://www.biorxiv.org/content/10.1101/2020.06.15.151779v1> In fact, the Cell's preview (entitled "Making Sense of Mutation: What D614G Means for the COVID-19 Pandemic Remains Unclear" <https://doi.org/10.1016/j.cell.2020.06.040>) for Korber's paper indeed cited our bioRxiv preprint. Still, unlike the Cell paper plus other bioRxiv papers that were posted with the same timing as ours, we did not focus only on the comparison between the prototype and D614G mutant of SARS-CoV-2. As described above in the response to Reviewer #1's comments#1, this present study is the first report to show the critical difference between SARS-S and SARS2-S in cell entry efficiency. Besides, to our knowledge, this is also the first description of a much stronger dependency of SARS2-S-mediated entry on TMPRSS2 expression than that of SARS-S-mediated entry.

It is very difficult for this reviewer to evaluate whether this study is novel enough because several papers on the same subject have been already published. Therefore, I left to the editor to make the final decision.

I have, however, the following 2 major points related to the revised version of the manuscript.

Point 1

3. Fig S2 This reviewer believes that a comparison with SARS in Fig S2C is missing to conclude that low rate cell entry is not due to differences in S incorporation into virions. Do the authors see same differences in entry when comparing 293T and lung cells or cells from the airway epithelial?

We appreciate the Reviewer's comment. This comparison is indeed our dilemma for the following reasons; The anti-S2 subunit monoclonal antibody (GeneTex, GTX632604) that we used in this study was originally generated against SARS-S, but not SARS2-S, so that the antibody can naturally react more efficiently with SARS-S than with SARS2-S. Even by using its tagged version using an anti-tag antibody, the SARS2-S incorporated into virions is completely cleaved by host cell proteases, whereas that of SARS-S is uncleaved and results in a two-times larger band size in immunoblot analyses (as reported by Hoffman et al. (Cell, 2020)), making it difficult to quantitatively compare these two different spike proteins. Instead of comparing them, we had chosen to compare the virion incorporation of SARS2-S WT and chimeric mutants that harbor SARS-S type of cytoplasmic tail (CT) or CT/transmembrane (TM), because accumulated evidence has shown that the compatibility of a variety of viral envelopes with lentiviral virions is determined by their CT domains. To emphasize this, we have added the following sentence with a couple of citations; "Nevertheless, ~25-fold differences in cell entry between SARS-S and SARS2-S pseudoviruses were observed (Fig. 1a), leading to the hypothesis that SARS2-S might be less compatible with lentiviral particles. Because a large body of literature has shown that lentiviral compatibility of a variety of viral envelopes is determined by their cytoplasmic tail (CT) domains¹²⁻¹⁴, we compared the sequences of these domains in S proteins and found that..." in the text (page 4, lines 9-15). Nevertheless, based on the fact that we did not directly and quantitatively compare the virion incorporation of these two spike proteins, we decided to change the wording of conclusions as follows (page 4, line 20 – page 5, line 1); "These results suggest that the significantly lower rate of cell entry of the SARS2-S pseudovirus was not due to the incompatibility

between SARS2-S and a lentiviral vector but rather to the intrinsic nature of the SARS2-S protein." Also, we added the following sentence "Note that SARS2-S incorporated into virions is completely cleaved by host cell proteases, whereas that of SARS-S is uncleaved and results in a two-times larger band size in immunoblot analyses (as reported by Hoffman et al. (Cell, 2020)), making it unable to directly compare these S proteins." in the legend of Extended Data Fig. 2.

As for the infection experiment using human airway cells, we have added the data in Extended Data Fig. 4., as described in the response to the comments#1

Answer

"Note that SARS2-S incorporated into virions is completely cleaved by host cell proteases, whereas that of SARS-S is uncleaved and results in a two-times larger band size in immunoblot analyses, making it unable to directly compare these S proteins".

This sentence was added to the Figure but the authors need to show the western blot that it is described in the sentence.

Key to the paper is to show that differences in pseudotyped virus entry are not due to differences in S incorporation into virions. Therefore, for this reviewer it is essential to show a direct comparison between SARS-CoV S and SARS-CoV-2 S incorporated in pseudotyped viruses. Using an antibody anti-tag for example and quantifying by western blot will provide a good approximation to the question. This reviewer does not agree with the authors that the different S processing state between the two S proteins from the two viruses is a problem to quantify and/or direct compare the spikes. SARS-CoV S will show one band (corresponding to the size of full S) and SARS-CoV 2 S will show a band corresponding to the size of S2, if 100% cleaved) and the intensity of the bands can be normalized to corresponding size. The comparison within Sars2 S variants is not addressing the question asked by this reviewer.

Point 2

4. Fig 2 Structural analyses on complex models between ACE2 and naturally occur S mutants. The authors hypothesized that the mutation D614G can provide flexible space between two protomers due to the short G side chain which might lead to improved accessibility of ACE2 into the RBD. To test this possibility, they performed in vitro binding and did not observe any difference in ACE2 binding between WT and D614G (Extended Data Fig. 4). If the amount of proteins use in the in vitro assay was above the Kd then is expected to see no differences. This reviewer thinks that the assay to do is to compare affinities between D614 and G 614 for ACE2 using octet or biacore. The sentence "Nevertheless, the aforementioned structural data are fully consistent with the high entry activity of the virus harboring this mutation (Fig. 1e)" should be remove since from the structural data is coming a hypothesis that actually the authors couldn't demonstrate.

"The assay to do is to compare affinities between D614 and G614 for ACE2 using octet or biacore" is unlikely for the following reasons: As reported by Shang et al. (PNAS, 117:11727-11734, 2020), their results obtained in IP-based comparative analysis of receptor binding affinity of SARS-S and SARS2-S proteins contradicted their own recent results obtained from Biacore based on surface plasmon resonance (SPR) by using purified RBD and ACE2 proteins (Nature, 581:221-224, 2020), and they therefore concluded that "whereas SARS-CoV-2 RBD has higher ACE2 binding affinity than SARS-CoV RBD, SARS2-S has lower ACE2 binding affinity than SARS-S". This contradiction is somewhat predictable because the interaction of purified RBD-ACE2 proteins they observed in Nature paper was an RBD monomer-vs-ACE2 dimer interaction that is different from their native binding conformation. For example, in the case of soluble versions of HIV-1 Envelope glycoprotein (Env), it consists of full-length gp120 and most of the gp41 ectodomain in which the cytoplasmic tail and transmembrane regions of gp41 are deleted. The natural association between gp120 and gp41 in the functional Env spike is non-covalent and labile, thereby recombinant expression of cleaved Env results in dissociation of the hetero-dimeric subunits (Sharma et al. Cell Rep. 11:539-550, 2015). Exactly the same thing can be said for S1 and S2 proteins of SARS-CoV-2 (which correspond to HIV-1 gp120 and gp41,

respectively).

For this reason, even a commercially available SARS2-S protein, such as a purified SARS2-S trimer (Acro Biosystems, SPN-C52H8), as well as the similar SARS2-S trimer just published by Yurkovetskiy et al. (Cell, <https://doi.org/10.1016/j.cell.2020.09.032> Sep. 15, 2020), has mutations in its cleavage site, resulting in non-cleaved precursor of SARS2-S that is stable but structurally different from the native form of this protein, which will make it difficult to precisely compare the ACE2 binding affinity of SARS-S and SARS2-S proteins, even by performing SPR-based Biacore.

Therefore, it would be technically difficult to reproduce the actual native binding between ACE2 dimer and either WT or D614G trimer by using purified proteins through either Biacore or Octet.

As suggested, the sentence "Nevertheless, the aforementioned structural data are fully consistent with the high entry activity of the virus harboring this mutation (Fig. 1e)" was accordingly removed.

Answer

The authors did not properly justify the request. Is it possible to measure affinities of recombinant spikes harboring 2P or 6P stabilized mutations, both designed by McLellan's lab.

Minor points

There are many parts of the manuscript that should be re-written like for example the following in the abstract:

"Among S variants, the D614G mutant shows the highest cell entry, as supported by structural observations. Nevertheless, the D614G mutant remains susceptible to neutralization by antisera against prototypic viruses. Taken together, these data indicate that the D614G mutation enhances viral infectivity while maintaining neutralization susceptibility"

It's basically saying twice the same thing.

Reviewer #2 (Remarks to the Author):

The revised manuscript by Ozono and co-authors has mostly addressed my initial concerns. In response to a concern related to the lack of direct comparison of the levels of pseudovirus incorporation of SARS-S and SARS-2 S proteins, the authors cite complications due to the difference in intracellular cleavage of SARS and SARS2 S proteins. While this is a fair point, I'd still show an immunoblot in a supplementary figure and let the readers judge the relative extents of S incorporation. I have no other concerns with the current version of the paper.

Reviewer #3 (Remarks to the Author):

While concerns about originality of the report remain, the authors have responded to previous reviews, and they have improved the manuscript with additional extended data.

Reviewer #1 (Remarks to the Author):

Comments

It is very difficult for this reviewer to evaluate whether this study is novel enough because several papers on the same subject have been already published. Therefore, I left to the editor to make the final decision.

I have, however, the following 2 major points related to the revised version of the manuscript.

Point 1

Key to the paper is to show that differences in pseudotyped virus entry are not due to differences in S incorporation into virions. Therefore, for this reviewer it is essential to show a direct comparison between SARS-CoV S and SARS-CoV-2 S incorporated in pseudotyped viruses. Using an antibody anti-tag for example and quantifying by western blot will provide a good approximation to the question. This reviewer does not agree with the authors that the different S processing state between the two S proteins from the two viruses is a problem to quantify and/or direct compare the spikes. SARS-CoV S will show one band (corresponding to the size of full S) and SARS-CoV 2 S will show a band corresponding to the size of S2, if 100% cleaved) and the intensity of the bands can be normalized to corresponding size. The comparison within Sars2 S variants is not addressing the question asked by this reviewer.

We thank the Reviewer#1 for this comment. Despite the size difference between SARS-S and SARS2-S proteins, we have newly created C-terminally-tagged S expression plasmids to avoid the problem of antibody affinity as suggested, and performed Western blotting assays by using them. We have added the new data in Extended Data Fig. 2 and the related sentence in the Methods.

Point 2

The authors did not properly justify the request. Is it possible to measure affinities of recombinant spikes harboring 2P or 6P stabilized mutations, both designed by McLellan's lab.

We deeply appreciate the Reviewer#1's comment. Thanks to the suggestion, we decided to utilize biolayer interferometry technology with Octet using highly purified 6P stabilized S proteins. We then finally successfully found the mechanism by which the D614G variant shows increased cell entry compared with WT. Therefore, we have replaced IP-Western data (ex-Supplementary Fig. 5) with the biolayer interferometry data (new Fig. 4), and added the related sentence into the revised manuscript.

Besides, based on this new finding providing the mechanism of D614G-mediated entry enhancement, together with the fact that four out of five main figures focus on analyses of the D614G variant, we would like to change the title from “Naturally mutated spike proteins of SARS-CoV-2 variants show differential levels of cell entry” to “SARS-CoV-2 D614G spike mutation increases entry efficiency with enhanced ACE2-binding affinity” under the permission of the Editor.

Minor points

There are many parts of the manuscript that should be re-written like for example the following in the abstract:

“Among S variants, the D614G mutant shows the highest cell entry, as supported by structural observations. Nevertheless, the D614G mutant remains susceptible to neutralization by antisera against prototypic viruses. Taken together, these data indicate that the D614G mutation enhances viral infectivity while maintaining neutralization susceptibility”

It's basically saying twice the same thing.

The first sentence is the description of results obtained from 1) cell entry assays using pseudoviruses, and structural modeling; 2) neutralization assays using sera from patients infected with the Wuhan prototype, and the second sentence is the conclusion by translating the results as; 1) S protein-based cell entry -> viral infectivity consistent with structural analysis; 2) Neutralization by prototype sera -> maintenance of neutralization susceptibility. Nevertheless, we agree that the current version sounds like simple repetition so that we have amended the sentence. Also, as suggested, we double-checked the manuscript, and revised some parts.

REVIEWERS' COMMENTS

Reviewer #1 (Remarks to the Author):

The authors answer my request performing a western blot to verify the S incorporation into virions to compare SARS-CoV and SARS-CoV 2 and therefore conclusively exclude the possibility that differences in cell entry are due to S incorporation into pseudotype virus.

This reviewer however, does not agree about the conclusions of the BLI experiment. To consider that two KD are different they should at least differ by 5 fold or to show statistically that they are different.

Reviewer #1 (Remarks to the Author):

The authors answer my request performing a western blot to verify the S incorporation into virions to compare SARS-CoV and SARS-CoV 2 and therefore conclusively exclude the possibility that differences in cell entry are due to S incorporation into pseudotype virus.

We thank the Reviewer#1 for this comment.

This reviewer however, does not agree about the conclusions of the BLI experiment. To consider that two KD are different they should at least differ by 5 fold or to show statistically that they are different.

As suggested, we provide a statistical data in Supplementary figure 5 showing that the difference in *KD* and *Kdis* was statistically significant.